# Glut 1 in Cancer Cells and the Inhibitory Action of Resveratrol as A Potential Therapeutic Strategy

**DOI:** 10.3390/ijms20133374

**Published:** 2019-07-09

**Authors:** Angara Zambrano, Matías Molt, Elena Uribe, Mónica Salas

**Affiliations:** 1Instituto de Bioquimica y Microbiologia, Universidad Austral de Chile, Valdivia 0000000, Chile; 2Departamento de Bioquímica y Biología Molecular, Facultad de Ciencias, Universidad de Concepción, Concepción 4070386, Chile

**Keywords:** GLUT1, glucose uptake inhibition, cancer therapy, cancer metabolism, resveratrol

## Abstract

An important hallmark in cancer cells is the increase in glucose uptake. GLUT1 is an important target in cancer treatment because cancer cells upregulate GLUT1, a membrane protein that facilitates the basal uptake of glucose in most cell types, to ensure the flux of sugar into metabolic pathways. The dysregulation of GLUT1 is associated with numerous disorders, including cancer and metabolic diseases. There are natural products emerging as a source for inhibitors of glucose uptake, and resveratrol is a molecule of natural origin with many properties that acts as antioxidant and antiproliferative in malignant cells. In the present review, we discuss how GLUT1 is involved in the general scheme of cancer cell metabolism, the mechanism of glucose transport, and the importance of GLUT1 structure to understand the inhibition process. Then, we review the current state-of-the-art of resveratrol and other natural products as GLUT1 inhibitors, focusing on those directed at treating different types of cancer. Targeting GLUT1 activity is a promising strategy for the development of drugs aimed at treating neoplastic growth.

## 1. Introduction

An important element in the glucose metabolism of cancer cells is the uptake of glucose. This process is well-regulated and involves several elements, such as the effect of signaling growth factors [1] and the interaction with the extracellular matrix [2,3]. To survive, cancer cells get oncogenic mutations to become independent of external regulation for glucose uptake [4,5].

Natural products have been an increasingly important source of drugs for cancer treatment. Resveratrol (RSV) is a natural compound present in red fruits with well-known antioxidant and antiproliferative properties [6,7]. RSV also has many targets in the cell and there are a lot of studies of the signal pathway implicated in the survival process [8,9]. Nevertheless, there are controversial reports of oxidant and antioxidant activities of RSV. This may be possibly explained by considering that RSV is a photosensitive molecule which must be carefully manipulated and used at the exact dose [10,11].

This review focuses on the importance of RSV and other small inhibitors having GLUT1 as the principal target in the cancer cell. In the first sections of the review, we illustrate the current vision of cancer cell metabolism and the role of GLUT1 as the limiting step in the flux of glucose. Then, we revisit different aspects of GLUT1, including kinetic, physiological, and some structural details. Finally, we summarize the numerous studies that have been focused on developing specific inhibitors of the activity of GLUT1, with special emphasis in natural compounds such as RSV.

There are many reports showing overexpression of GLUT1 in cancer cells [12,13,14,15]. Clinically, there is a correlation between glucose uptake, diagnosis, and prognosis of tumors [16,17]. We will also describe the mechanism involved in glucose transport, which is an important step for glucose uptake.

We also consider and define the differences in the energy metabolism of normal and cancer cells, as a biochemical basis for the identification of possible targets for metabolic therapy in the treatment of cancer. A promising strategy for antitumor therapy could be the inhibition of glucose transporters in neoplastic cells, thus generating a state of energy deprivation that may facilitate the effect of other anticancer therapies [18,19]. It is interesting to investigate the possibility of using natural drugs such as RSV in combinatorial treatment in order to reduce the side effects and improve cancer treatments.

## 2. Glucose Uptake and Cancer Metabolism

In order to satisfy the higher demands of metabolites necessary for an increased proliferation, the cancer cell needs to improve the nutrients uptake from the medium. Glucose is the main nutrient for proliferation in eukaryotic cells. By means of glucose metabolism, the cell generates various intermediates and also generates reduced equivalents, such as NADH, FADH_2_, and NADPH, which are necessary for anabolic reactions in order to maintain a high growth rate (Figure 1). The first step of glycolysis is glucose phosphorylation by hexokinase (HK), and after many reactions pyruvate, NADH and ATP are finally obtained. In the presence of oxygen, normal cells convert pyruvate to Acetyl-CoA and use the Krebs cycle and oxidative phosphorylation (OXPHOS) pathways, which produce more ATP than glucose fermentation [20]. In contrast, cancer cells are highly dependent of the glycolytic pathway [21,22] to meet their energy requirements and prefer glucose fermentation over mitochondrial oxidation, even under aerobic conditions; this is called the Warburg effect [23] (Figure 1, blue line). In many cancer cells, there is also an upregulation of the hexokinase 2 enzyme (HK2) [24,25], together with an increased conversion of pyruvate to lactate (fermentation), which regenerates the NAD^+^ that feeds glycolysis, and also shows less OXPHOS activity, even in presence of oxygen. In this connection, Fantin et al. in 2006 observed that suppression of aerobic lactate production increases the OXPHOS activity in mammary tumor cells [26], clearly showing the preference of cancer cells for glycolysis. On the other hand, Shim et al. in 1998 found that apoptosis could be caused by glucose deprivation of transformed cancer cells [27], thus reinforcing the importance of the Warburg effect in cancer proliferation. 

However, there are also tumors that use OXPHOS as a predominant ATP-generating mechanism [28,29,30,31,32] (Figure 1, red line). In fact, it has been demonstrated that many tumor cell lines, such as U-937 and HeLa cells, used mitochondrial respiration to support their growth [29,33,34,35,36]. The detection of higher mitochondrial membrane potential in tumors than normal cells provides an advantage for the detection of oxidative tumors [37]. On the other hand, some malignant cells utilize a metabolic strategy involving aerobic glycolysis and increased activity of the Krebs cycle, fed by anaplerotic reactions with carbons from fatty acids, glutamate, glutamine or aspartate. In this connection, there are reports showing that inhibition of ATP synthesis by mitochondrial uncoupling agents in cancer cells, results in increased production of lactate in the presence of oxygen [38,39] (Figure 1, black line). Interestingly, these cancer cells became resistant to apoptosis [40]. With regard to mitochondrial uncoupling, there are mitochondrial carriers named uncoupling proteins (UCPs) that are able to control mitochondrial ROS generation that favors the survival of cancer cells [41]. UCPs have been connected with proliferation and anaplerotic mitochondrial metabolism in cancer cells. In agreement with this, Esteves et al. in 2015 used microarray analysis to report the existence of 2 different types of cancer on the basis of UCP2 mRNA expression level with probably some metabolic differences, opening questions related to glycolysis, OXPHOS, and proliferation that must be elucidated [42]. 

Independently of which strategy is followed by a cancer cell, a hallmark of cancer is the increase in glucose uptake, and taking advantage of this fact, the positron emission tomography (PET) has been used for cancer diagnostic with the glucose analog 2-(18F)-fluoro-2-deoxy-D-glucose (FDG) as a tracer of glucose uptake [43]. Nevertheless, this technique does not permit to evaluate whether glucose is utilized specifically by glycolysis or by the aerobic pathway including OXPHOS.

There are different studies that indicate that the Warburg effect and the mitochondrial function occur simultaneously [44]. Therefore, the proliferating cells may use one or both ATP-producing pathways, and an understanding of the limiting reactions in their metabolism will be useful for improving anticancer therapies.

## 3. Glucose Uptake and Glucose Transporters

### 3.1. Glucose Transporters

Since sugar does not simply diffuse across the lipid bilayer, glucose uptake may be considered as the rate-limiting step for cancer cell metabolism and growth. Understanding the molecular mechanism of glucose transport becomes highly relevant because it is the starting point for the rational design of drugs aimed at blocking sugar uptake, which translates into stalling cell growth and eventually tumor death.

Glucose crosses the lipid bilayer through specialized membrane proteins known as sugar transporters. There are three well-known families of sugar transporters in mammals: GLUT family (glucose transporters), SGLT family (sodium-glucose linked transporters), and the recently discovered SWEET family (sugars will eventually be exported transporters) [45,46,47,48]. While describing SGLT and SWEET carriers is out of the scope of this review, here we briefly summarize their main features. 

SGLT transporters are symporters that move glucose against a concentration gradient by coupling the cotransport of sodium down its concentration gradient. Thus, SGLT carriers are energetically linked to a secondary transport system, Na^+^/K^+^ ATPase, which maintains the Na^+^ gradient. This sets the main difference with GLUT transporters, which facilitate the movement of glucose down the concentration gradient without requiring primary or secondary ATP hydrolysis.

SGLT carriers are essential for glucose absorption in the small intestine and glucose reabsorption in the renal tissue (Further information about the function of SGLT transporters can be found in references [46,49]). The SWEET carriers are less understood, although they may be part of a sugar export system. In plants, SWEET carriers play a role in the efflux of glucose of several physiological processes [46,47], including seed filling [50] and nectar secretion [51]. In mammals, SWEET transporters are localized in the Golgi and catalyze the efflux of glucose from Xenopus oocytes [52], suggesting its participation in glucose export [47].

Regarding the GLUT family, there are 14 members that differ in substrate selectivity, tissue expression and subcellular localization [45,53]. These carriers can be further classified into three classes based on substrate specificity and sequence similarity: Class I, which includes GLUT1-4 and 14, they have high selectivity for glucose; class II, comprising GLUT5, 7, 9, and 11, which have selectivity for glucose and fructose; and the uncharacterized class III, which include GLUT6, 8, 10, 12, and 13 (known as HMIT). Members of class I are the best understood as they directly regulate sugar homeostasis. GLUT1 is expressed in most cell types and has a *K_M_* for glucose of about 3–7 mM [45,54,55]. GLUT2, 3, and 4 are also well-studied proteins because of their relevance, together with GLUT1 in sugar homeostasis. GLUT2 and GLUT3 are low and high-affinity glucose carriers with *K_M_* for glucose of about 17 and 1 mM, respectively. GLUT2 is localized in the small intestine, liver, and pancreatic tissue, while GLUT3 is mainly found in the brain. GLUT4 has a *K_M_* for glucose similar to that of GLUT1 [45,54] and is localized in vesicles that fuse with the plasma membrane in response to insulin signaling. Any dramatic and steady increase in glucose concentration, e.g., after a meal rich in sugars, induces translocation of GLUT4 to the plasma membrane [56], which lowers blood levels of glucose. Consequently, the physiological extracellular glucose levels are kept around 5 mM because of the direct action of GLUT1 and GLUT4. A detailed review of the GLUT family can be found in Reference [45].

### 3.2. GLUT1: Kinetic Properties and Mechanism of Transport

GLUTs proteins must perform repeated cycles of conformational changes to load and release substrate molecules on opposite sides of the biological membrane. The transport process of the GLUTs that involucrate multiple conformational changes is an essential first step for elucidating the transport mechanism. The resolution of structures of human GLUT 1 and 3 and rat GLUT 5, in different conformational states, have served as a framework for the understanding of their functional mechanism [48,57,58,59,60]. The GLUT 1 transporter carries out the typical alternative access mechanism for the translocation of a substrate [61], which involves the state open to the outside, closed and open towards the interior. The substrate-binding site is alternatively shown on each side of the membrane and structural changes are continuous during a transport cycle. The comparison of GLUT1/3/5 structures in different conformational states indicates that the entry of the substrate causes movements of domains and local conformational adjustments [48,58,59,60,61]. The conformational change from the outside to the inside of the transporter implies the rotation of the N and C domains at ~15 degrees [61]. The N domain remains practically unchanged during conformational changes, suggesting a rigid rotation with respect to C domain. In contrast, in the C-terminal domain, the discontinuous TM7/10 helices and their neighboring segments suffer prominent local rearrangements. TM7b leans towards the core of the cavity, and TM10b moves away from the transport path while the substrate is released [58]. The difference in the conformational changes of the N and C domains would be due to a different composition of amino acids and the structural properties of each domain. The internal core of N domain is mainly hydrophilic, although the surface oriented to the substrate is largely hydrophobic. Seven water molecules located within the N domain form H bonds with the polar residues of the TM1 and TM4 in the internal core of the N domain, which may maintain the rigidity of the N domain during the conformational changes [57]. In contrast, the internal core of the C domain is highly hydrophobic and contains only one molecule of water, although the substrate-oriented region contains polar residues [61,62]. Furthermore, GLUT proteins have an intracellular domain (ICH) that is formed by three or four helices located between the N and C domains and a short helix at the C-terminus [63]. Together with the TM domains, the ICH domain also undergoes important changes during the translocation of the substrate. This generates a rearrangement of ICH interactions with the N and C domains. It is proposed that the ICH domain functions as a “door lock” that ensures the intracellular gate, also stabilizing the external conformation [57]. In the same way, Quistgaard et al. in 2016 [64] proposed a model for the MFS family of transporters that considers a movement from an occluded state to a rotation of N- and C-terminal domains outward facing or inward facing, or vice versa, that facilitates the movement of substrate from outside to inside of the membrane and vice versa during the transport cycle, reinforcing the alternating access mechanism of GLUTs. The structural information has been fundamental to elucidate the transport mechanism of GLUTs and supports the design of drugs to new therapies against several human diseases such as cancer.

### 3.3. GLUT1 Overexpression in Cancer Cells

As we mentioned before, GLUT1 expression is an important hallmark in many types of cancers, [12] including breast cancer, gastric adenocarcinoma, and many other types of cancer (see Table 1). Interestingly, some cancers have been reported with a normal o decreased expression in GLUT1, such as Sarcomas, lymphomas, melanomas, and hepatoblastomas [65,66]. It is important to mention that Oh et al. in 2017 [67] made a downregulation of GLUT1 expression in breast cancer cell, getting a decrease in cellular apoptosis, induced by an up-regulation in Akt signaling pathway, suggesting that drugs focused only on GLUT1 as a therapeutic target must be evaluated in each cancer cell type. 

Table 1 lists several pieces of evidence indicating the overexpression of GLUT1 in various cancer models and cell lines.

### 3.4. How Is GLUT1 Regulated?

Until now, several meta-analyses have suggested that GLUT1 could be used as an optimal biomarker in several cancer types. Indeed, the overexpression of GLUT1 is correlated with prognosis and survival in solid tumors [94,95].

In general, GLUTs are regulated by several molecular mechanisms. Specifically, the expression of GLUT1 is regulated by many transcription factors. It has been demonstrated that hypoxia-inducible factor (HIF-1alpha) promotes GLUT1 expression in hypoxic conditions [96,97,98], c-Myc has been shown to regulate GLUT1 expression in many tumors [99], and the un-controlled Ras pathway induces the up-regulation of GLUT1 expression [100,101]. The anomalous expression of GLUT1 is also specifically affected by the PI3K/Akt pathway [102,103]; Akt has been involved in the expression of GLUT1 and GLUT3 in cancer cells [104]. 

GLUT1 mRNA have different sequence motifs within the 3’-UTR, which are involved in the transcript stability and control of GLUT1 expression [105]. The alteration of the stability of GLUT1 transcription is associated with a variation of glucose concentrations, and the presence of growth factors, cytokines, and some hormones. Also, it has been demonstrated that the long non-coding RNA HOX transcript antisense RNA (HOTAIR) is able to up-regulate GLUT1 [106], and the same goes for the miRNA-150, which is able to regulate glycolysis by modifying GLUT1 expression [107].

The presence of growth factors is involved not only in modifying gene expression, but they also have further an important role inducing the translocation from intracellular compartments to the plasma membrane [108]. Many of these growth factors stimulate PI3K/Akt signaling pathway, which enhances GLUT1 activity by increasing the membrane trafficking [103,109,110]. This pathway is known as a master regulator of cell growth and proliferation and is highly mutated in human cancers.

## 4. Inhibition of Glucose Uptake by Resveratrol

Resveratrol (3,5,4′-trihydroxystilbene or RSV) is a polyphenolic natural product that attracted great interest mainly due to its anticarcinogenic, anti-inflammatory, and cardioprotective properties [111,112,113]. RSV has structural similarities with tyrosine kinases that are known inhibitors of GLUT1 [114,115]. For this reason, we performed experiments in order to investigate the relationship between RSV and this transporter. By using kinetic assays, we observed for the first time that RSV inhibit the glucose uptake in human leukemic cell lines U-937 and HL-60, by a direct interaction with the internal face of GLUT1, in a noncompetitive mode [116].

With regard to RSV and glucose uptake, studies in different human ovarian cancer cells have shown that treatments with RSV were able to inhibit glucose uptake, lactate production, Akt, and mTOR signaling, and cell viability depending on the dose and time used [117,118,119]. There are a great number of studies relating RSV and glucose uptake in cancer cells and in pathological conditions such as insulin resistance that we summarize in Table 2. Most of the experiments were done in vitro, but also in vivo using labeled glucose analogs. In cancer cells it is observed that resveratrol inhibited the uptake of glucose, favoring the anticancer action; but in pathological conditions such as insulin resistance or diabetes, resveratrol increased glucose uptake and insulin sensitivity favoring the antidiabetic effect [120]. Other studies in neuronal cells showing that RSV had an inhibitory effect of the glucose uptake, favoring the neuronal regulation of glucose and insulin sensitivity [121]. Among other mechanisms of action RSV also targets a great number of intracellular molecules implicated in cell cycle control and apoptosis induction [122,123,124,125,126]. RSV is an attractive candidate for cancer therapy because of its unique capacity to affect the mTOR/AMPK pathway at different levels. By inhibiting mTOR and ribosomal protein S6 kinase, and by activating AMPK, RSV exerts a potent short-term effect on metabolism. The type of cell death observed in cancer cells treated with RSV has been reported as apoptosis or autophagy [118,119]. Van Ginkel et al. [127] concluded that elevated levels of RSV lead to tumor regression and widespread tumor cell death. The underlying mechanism involves direct activation of the intrinsic and extrinsic apoptotic pathway. Thus, in normal adipocytes, it has been observed that RSV induces apoptosis at concentrations greater than 20 µM, while in insulin-resistant adipocytes; RSV stimulates glucose transport via SIRT1-AMPK-Akt. These results suggest that RSV can behave differently according to the dose used and the cell type and the metabolic state [120]. Recently, Dai et al. [128] showed that RSV inhibits the growth and proliferation of MGC-803 cells of gastric cancer in a dose- and time-dependent way by downregulating the expression of genes related with Wnt pathway as β-catenin, c-myc, and cyclin D1, proteins related with GLUTs gene regulation. On the other hand, Kleszcz, et al. [129] did not observe inhibition of c-myc gene expression by resveratrol in FaDu hypopharyngeal carcinoma cells, but the doses used were significantly lower. With respect to autophagy, a number of studies show that RSV induces autophagy and cell death in cancer cells when they are nutrient-deprived and that RSV could act by inducing a starvation-like signaling response [117]. Indeed, activation of the JNK pathway by RSV leads to the induction of genes that participate in both the initial and late steps of autophagy in CML cells [130,131]. RSV could also impact mitochondrial membrane potential, the respiration chain, and ATP synthesis [132].

RSV also stimulates the molecular pathway dependent on sirtuins, histone deacetylases, that regulates the activity of transcription factors in many tissues related to energy metabolism [133]. RSV induces the expression of silent information regulator-6 (SIRT6) in hypopharyngeal carcinoma FaDu cell line [129]. SIRT6 influences the expression of several glycolytic genes such as GLUT1, aldolase, pyruvate dehydrogenase kinase 1 (PDK1), and phosphofructokinase 1 (PFK1) [134]. Furthermore, RSV is known as an activator of SIRT1 and has been related to diabetes progress as a target for its treatment [135,136], inflammation, and neuroprotection [137]. There are data suggesting that RSV has hypoglycemic properties in diabetic rats and restore glycolytic enzyme activities [138,139]. 

With respect to the repertoire of glycolytic isoenzymes involved in the action of RSV, Iqbal et al. observed that RSV down-regulate pyruvate kinase 2 (PKM2) expression by inhibiting mTOR signaling and suppressed cancer metabolism, characterized by a decreased glucose uptake, lower lactate production (aerobic glycolysis), and reduced anabolism (macromolecule synthesis) in various cancer cell lines [140]. Recently, it was observed that RSV, by targeting PKM2 and ERK1/2, destabilizes BCL-2 protein level finally leading to apoptosis in human melanoma cells [141]. In addition, the re-expression of the embryonic isoenzyme M2 of pyruvate kinase in cancerous cells has been related to a stronger glycolytic phenotype and a proliferative advantage in hypoxic conditions [142]. HK2 seems to be associated with mitochondria, linking ATP production in this organelle to cytosolic glucose phosphorylation. The release of mitochondrial HK2 could explain the increase in cytosolic hexokinase activity implicated in the onset of apoptosis [143]. In cardiomyocytes, during anoxia/deoxygenation injury, RSV exerts a protective effect by promoting the linkage of voltage-dependent anion channel 1 (VDAC1) to HK2 [144]. HK2 links up with VDAC1 forming a polymeric channel that finally stimulates cell survival [145].

Recently, RSV showed a decrease in mRNA and protein levels of GLUT1, HK2, PFK1, and PKM2 which finally caused inhibition of aerobic glycolysis in a study of VEGF-angiogenesis in human umbilical vein endothelial cells, putting forth the role of RSV in the regulation of pathological angiogenesis [146].

Specifically, RSV exerts effects on the GLUT1 transporter at different levels, such as:

(a) A direct inhibitory action on the protein, which has been demonstrated by kinetic assays on cancer cells (116) and interrupting the traffic to the plasma membrane [147].

(b) An inhibitory effect on mRNA expression for GLUT [145,148,149,150]

(c) Regulating many transcription factors that in turn regulate the expression of GLUT1 such as HIF-1alpha and c-Myc [129,151]

(d) Regulating GLUT1 expression through various signaling pathways such as: AMPK [147], Wnt [127,128], Jnk kinases [129,130], sirtuins and histone deacetylases [132].

(e) Also regulating miRNA expression of GLUT1.

RSV also is able to regulate glucose uptake, metabolism and signaling pathway in cancer cells through regulation of specific microRNAs (miRNAs). Resveratrol affects the miRNA machinery in positive and negative manners; it is suggested that this regulatory activity is likely to be advantageous for cancer treatment and prevention. There are many miRNAs that are dysregulated in cancers [152,153,154,155,156,157]. In resveratrol-treated prostate cancer cells there are significant upregulations of 28 miRNAs and downregulations of 23 miRNAs. Among these, two miRNA clusters, such as miR-17-92 and miR-106ab, are known oncomirs. Subsequent analyses showed significant downregulation of these oncomirs in resveratrol-treated prostate cancer cells [158]. In the case of breast cancer, miR-663 and miR-744 have been found to negatively regulate eEF1A2, resveratrol induces a 4.5-fold upregulation of miR-663 and a two-fold increase in miR-744 [152], resveratrol also controls breast cancer cell proliferation by inducing tumor-suppressive miRNAs (miR-34a, miR-424, and miR-503) via the p53 pathway and then by suppressing heterogeneous nuclear ribonucleoprotein A1 (HNRNPA1), which is associated with tumor progression [159]. There are many interesting studies about the molecular mechanisms associating resveratrol and miRNA regulation in cancer. For example, resveratrol targets the oncogenic expression of miR-21, thus blocking the PI3K-Akt signaling [160], and it also reduces MMP2 via upregulation of miR-328 in osteosarcoma cells [161].

Several miRNAs have an important impact on metabolism in cancer cells. Some miRNAs such as miR-150 were consistently decreased in cell lines and osteosarcoma tissues as compared to osteoblast cells and normal bone; the ectopic overexpression of miR-150 inhibits osteosarcoma cell proliferation and suppresses glucose uptake. On the other hand, loss of function of miR-150 enhanced osteosarcoma cell proliferation and increased glucose uptake and lactate secretion [107]. In colon cancer cells, the miRNA-143 overexpression inhibits glucose uptake and glucose transporter 1 (GLUT1) expression [162]. Many miRNAs are associated to regulated GLUTs expression, for example; miR-21a-5p, miR-29a-3p, miR-29c-3p, miR-93-5p, miR-106b-5p, miR-133a-3p, miR-133b-3p, miR-222-3p, and miR-223-3p have been reported to directly and/or indirectly regulate the GLUT4 expression; and their expressions are altered principally in the diabetes condition [163]. 

Recent evidences demonstrate that miRNAs play important roles in certain effects of resveratrol on cell metabolism. Pyruvate kinase M2 (PKM2) has been found to be overexpressed in different cancers [164]; it has been shown that resveratrol represses PKM2 by increasing the expression of miR-326. Also, resveratrol improves mitochondrial function. Specifically, the miR-27b was significantly induced in a dose-dependent way in skeletal muscle and C2C12 myoblast treated with resveratrol, and miR-27b overexpression improves mitochondrial function in a Sirt1-dependent manner [165].

## 5. Inhibition of GLUT1 by Other Small Molecules

Because of the relevance of GLUT1 in cancer, GLUT1 is a potential target for therapy with small molecule inhibitors [169]. GLUT1 is inhibited by small natural and synthetic molecules belonging to different families of organic compounds. Table 3 lists different small molecule inhibitors that have been characterized by testing natural compounds or libraries of preexisting compounds. Several of these molecules can be obtained directly from the diet, as they are found in fruits and leaves (e.g., grapes), and bind directly to the transporter. Examples include phloretin and cytochalasin B [170], flavones and flavonoids [114,171], and tyrosine-kinase inhibitors [114,171]. Also, we have characterized the effect of NDGA [172], gossypol [173], methylxanthines [55], and resveratrol [116] in glucose uptake using kinetic analyses. These natural compounds showed different patterns of inhibition, including acting as competitive, noncompetitive and mixed inhibitors. Although these kinetic analyses provide evidence on the effects of small molecules on GLUT1, little is known about the structural or functional details behind the inhibition. For example, the exact binding sites are unknown, although the current knowledge points to its binding to the extracellular or intracellular face of the transporter. Even in the case of competitive inhibition, it is not clear whether these small molecules bind to the glucose binding site or to a second site that makes glucose or inhibitor binding mutually exclusive, or both.

## 6. Conclusion and Final Remarks

Natural products have been the inspiration and the source of drugs and medicines that we currently use. RSV, due to its hypoglycemic and inhibitory of glucose uptake properties, emerged as an alternative for different diseases such as diabetes and cancer [176]. Throughout this manuscript, we have reviewed the importance of RSV and GLUT1 in glucose metabolism since metabolic alterations are a hallmark of cancer. Also, we have described everything related to the structure, regulation, expression, and overexpression of GLUT1 used as diagnostic for cancer. All this important information has served as a basis for the design of transport inhibitory molecules in different target cells. The latest structural data made it possible to design in silico specific inhibitors of GLUTs, to be validated in vitro and in vivo. Transport experiments in cells are difficult, which makes this goal tough. While all cells need glucose to live, partial inhibition of transport and specifically GLUT1 as mono-therapy was unsuccessful. New and novel combinatorial strategies that could use GLUT1 inhibitors such as RSV with anticancer conventional drugs for therapy are promising, decreasing the side effects of and maximizing the therapeutic effects.

## Figures and Tables

**Figure 1 ijms-20-03374-f001:**
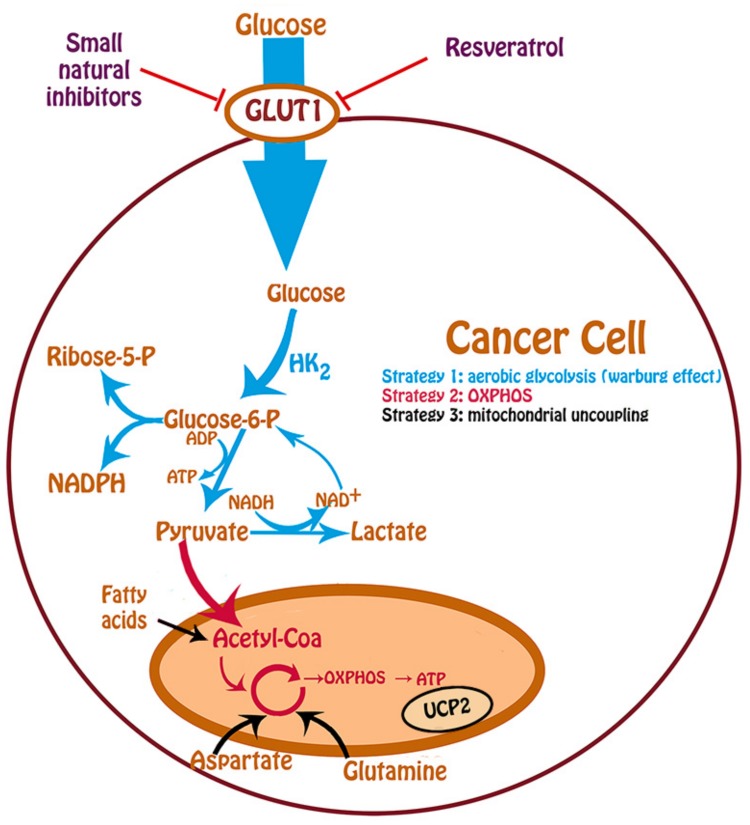
Glucose metabolism in cancer cells. In strategy 1, glucose is uptake is by GLUT1, which is a glucose transporter overexpressed in cancer cells, and follows the glycolytic pathway to pyruvate. The first step is the phosphorylation by hexokinase (HK) and the main isoform upregulated in cancer is HK_2_. Under anaerobiosis, pyruvate is converted to lactate, with the regeneration of NAD^+^ that feeds glycolysis. This way is a source of energy for cancer cells and supplies intermediates like ribose-5-phosphate and NADPH that are required for cell proliferation. In strategy 2, pyruvate obtained by glycolysis is transformed in Acetyl-Coenzyme-A (Acetyl-CoA) that enters in the mitochondrial Krebs cycle and follows oxidative phosphorylation (OXPHOS), the main source of ATP in a normal cell. Finally, in strategy 3, the cancer cell may turn in mitochondrial uncoupling, because it could use substrates that are different of glucose as a carbon source like fatty acids, aspartate, and glutamine that feeds the Krebs cycle (anaplerotic reactions). There is a mitochondrial carrier named uncoupling protein 2 (UCP2), that has been connected with proliferation and anaplerotic mitochondrial metabolism in cancer cells. The proliferating cells can choose more than one strategy at a time.

**Table 1 ijms-20-03374-t001:** Overexpression of GLUT1 in human cancers.

Cancer Type	References
Breast cancer carcinoma and adenocarcinoma	[14,65,68]
Ovarian carcinoma	[15,69,70]
Prostate carcinoma and adenocarcinoma	[65,71,72]
Thyroid carcinoma and adenocarcinoma	[65,73]
Gastric adenocarcinoma	[65]
Rectal carcinoma	[74]
Squamous cell carcinoma of the head and neck	[65,75]
Uterine cervix squamous cell carcinomas	[65]
Glioblastomas	[65]
Retinoblastomas	[65]
Colorectal carcinoma and adenocarcinomas.	[76,77,78]
Nonsmall cell lung carcinoma	[68,79]
Oral squamous cell carcinoma	[80,81]
Squamous cell carcinoma of the tongue	[82]
Esophageal cancer	[83]
Urothelial papilloma	[84]
Meningioma	[85]
Brain tumors	[13]
Laryngeal carcinoma	[86]
Nasopharyngeal diffuse large b-cell lymphoma	[87]
Pancreatic neoplasia	[88]
Renal cell carcinoma	[89,90]
Hepatocellular carcinoma	[91]
Lung cancer	[92]
Cervical cancer	[93]

**Table 2 ijms-20-03374-t002:** Resveratrol and glucose uptake.

Glucose Uptake	Glucose Analog Used	Cell Type	Reference
inhibit (in vivo)	2-deoxy-2-[18F]fludeoxyglucose ([^18^F]FDG) uptake	A2780, SKOV3 (injected in female nu/nu mice).	[166]
inhibit (in vivo)	2-deoxy-2-[^18^F]fludeoxyglucose ([^18^F]FDG) uptake	LLC (injected in BALB/c-n mice)	[151]
inhibit (in vitro)	3-O-methyl-D-glucose (OMG) or 2-deoxy glucose (2-DG) uptake	HL60, U937, RBC	[116]
inhibit (in vitro)	(2-[N-(7-nitrobenz-2-oxa-1,3-diazol-4-yl)amino]-2-deoxyglucose) (2-NBDG) uptake	PA-1, OVCAR3, MDAH2774	[147]
inhibit (in vitro)	[^3^H]-2-DG uptake	A2780, MDAH-2774, HOC-1, HOC-8, OVCA 429, and OVCA 432 SKOV3	[117]
inhibit (in vitro)	Glucose Oxidase Assay Kit	A2780, SKOV3	[119]
Increase (in vitro)	2-deoxy-D-[^3^H] glucose uptake	3T3-L1	[120]
Inhibit (in vitro)	Glucose (hexokinase) assay kit	HUVEC	[149]
inhibit (in vitro)	(2-[N-(7-nitrobenz-2-oxa-1,3-diazol-4-yl)amino]-2-deoxyglucose) (2-NBDG) uptake	Neuro-2a (N2A)	[121]
Increase (in vitro)	2-doxy-D-glucose (2DG)	L6	[167]
Increase (in vitro)	[^3^H] DG	BeWo	[168]
Increase (in vitro)	2-deoxy-D-glucose (2DG)	Placental lobules	[150]

**Table 3 ijms-20-03374-t003:** Molecule inhibitors of GLUT1.

Inhibitor	IC_50_ (Ki), µM *	Type of Inhibition	Cell Type	Reference
Flavones and Isoflavones				
Genistein	10–15 mM (4–15)	competitive	HL60, CHO, RBC	[114,115,171]
Myricetin	(23)	competitive	HL60, CHO, RBC	[114]
Quercetin	(8–16)	competitive	HL60, CHO, RBC	[114,171]
Morin	(105)	competitive	HL60, CHO, RBC	[114]
Rhamnetin	(20)	competitive	HL60, CHO, RBC	[114]
Isorhamnetin	(5)	competitive	HL60, CHO, RBC	[114]
Biochanin A	(17)	competitive	HL60, CHO, RBC	[114]
**Lavendustin and Tyrphostins**				
Lavendustin A	(10)	competitive	HL60, CHO, RBC	[114]
Lavendustin B	(15)	competitive	HL60, CHO, RBC	[114]
Tyrphostin B44	(90)	competitive	HL60, CHO, RBC	[114]
Tyrphostin B46	(20–45)	competitive	HL60, CHO, RBC	[114,171]
Tyrphostin B48	(50)	competitive	HL60, CHO, RBC	[114]
Tyrphostin B50	(45)	competitive	HL60, CHO, RBC	[114]
Tyrphostin B56	(170)	competitive	HL60, CHO, RBC	[114]
Tyrphostin AG879	(85)	competitive	HL60, CHO, RBC	[114]
Tyrphostin A47	(115–160)	noncompetitive	HL60, CHO, RBC	[114,171]
**Other Tyrosine Kinase Inhibitors**				
Methyl 2,5- dihydroxycinnamate	(150)	noncompetitive	HL60, CHO, RBC	[114]
Gossypol	30 (7)	noncompetitive	HL60, CHO, RBC	[173]
Methylxanthines				
Pentoxifylline	4.7 mM (2.8)	uncompetitive	RBC	[55]
Caffeine	10 mM (4.5)	uncompetitive	RBC	[55]
Theophylline	14.4 mM (5.3)	uncompetitive	RBC	[55]
Phloretin	40		RBC	[55]
**Other Polyphenols**				
Resveratrol	30 (122)	noncompetitive	HL60, U937, RBC	[116]
NDGA	53–85 mM (4.5)	noncompetitive	HL60, U937, RBC	[172]
Gossypol	30 (7)	noncompetitive	HL60, CHO, RBC	[173]
Kaempferol	4	mixed	MCF-7	[174]
Curcumin	19	mixed	L929	[175]

* IC_50_: correspond to 50% of total inhibition in viability experiments; *K_i_*: correspond to inhibition constant in transport experiments.

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
