# Peer review of "Glut 1 in Cancer Cells and the Inhibitory Action of Resveratrol as A Potential Therapeutic Strategy"

_ijms, 2019, doi:10.3390/ijms20133374_

Round 1

Reviewer 1 Report

1. This study is aim to the regulation of GLUT in cancer cell metabolism. However, the subtitle of this review is '' resveratrol as an inhibitor''. In this review, author should provide more information about the regulation between resveratrol and GLUT. It will be better to summarize the evidence of resveratrol as an inhibitor of GLUT as a table. In addition, author should integrate the role of resveratrol on GLUT modulation by a figure.

2. Author also mention miRNA is able to regulate GLUT expression, thus, is there any role of resveratrol on miRNA? Is it possible to summarize several important miRNA which plays role in GLUT expression and how resveratrol interact with them.

3. The difference of the regulation potential between resveratrol and other small molecules should be discussed.

Minor:

Line 158, 5 and mM need to be added with space.

Author Response

1. In agreement with the reviewer, we have changed the title of the manuscript,before the title was “GLUT 1 IN CANCER CELLS: Resveratrol as an inhibitor”, now the title is “GLUT 1 IN CANCER CELLS AND THE INHIBITORY ACTION OF RESVERATROL AS A POTENTIAL THERAPEUTIC STRATEGY.”

Also,  we have incorporated the Table 2 in section 4, specifically with the information about the relation between resveratrol and glucose uptake. Finally, in the figure 1, we have incorporated the role of resveratrol on glucose uptake for the better understanding.

2. In Section 4, we have incorporated a paragraph with information about the role of resveratrol on miRNA in cancer, specifically the effect of resveratrol on GLUT1.

3. Section 5 describes the inhibitory effect of small molecules on GLUT1, including resveratrol. The effects are similar and are directly associated to glucose transport mediated by GLUT1, and the type of inhibition, as a general and common mechanism for all these molecules. The particular effect of each molecule is described in table 3.

4. The space has been added.

Reviewer 2 Report

1. Based on its title, the review is supposed to focus on the role of GLUT1 in cancer cell growth and survival and how it can be targeted for cancer treatment by the natural stilbenoid resveratrol. However, the manuscript reviewed in detail on many other aspects, such as the glucose metabolism pathways in cancer cells, different glucose transporters, the kinetic property and working mechanism of GLUT1 in mammalian cells, and the GLUT1 inhibition activity of other, non-resveratrol molecules.

2. More review on GLUT 1 in cancer cells can be added. In the manuscript, only section 3.3 covered this topic, and it only described the over-expression of GLUT 1 in different cancer cells. Although some cancer-specific regulation mechanisms were mentioned in 3.4, more information regarding the role of GLUT 1 in cancer cells can be included since this should be one of the primary focus of this review.

3. Some of the anti-cancer studies on resveratrol that were discussed in this review are in fact not directly related to GLUT1 or glucose metabolism.

Author Response

1. In section 4, pertinent information has been incorporated in relation to the action of resveratrol on the uptake, metabolism and signaling of glucose in cancer.

2. We have incorporated more information in section 4, including a new table that links the uptake of glucose in cancer cells and the effect of resveratrol.

3. Resveratrol has multiple targets within the cell. In section 4 we add information that relates the effect of resveratrol on the uptake of glucose in cancer. 

Round 2

Reviewer 1 Report

The authors have addressed my concerns and revised their data accordingly. The current manuscipt has been significantly improved. I do not have further questions regarding the current manuscript.

Reviewer 2 Report

In the revised manuscript the authors have properly addressed the initial comments and concerns.